# Binder-Free MnO_2_/MWCNT/Al Electrodes for Supercapacitors

**DOI:** 10.3390/nano12172922

**Published:** 2022-08-24

**Authors:** Arkady N. Redkin, Alena A. Mitina, Eugene E. Yakimov

**Affiliations:** Institute of Microelectronics Technology and High-Purity Materials, Russian Academy of Science (IMT RAS), Moscow District, 6 Academician Ossipyan Str., 142432 Chernogolovka, Russia

**Keywords:** supercapacitors, carbon nanotubes on aluminum foil, binder-free electrodes, MnO_2_/CNT composite, pseudocapacitance

## Abstract

Recently, significant progress has been made in the performance of supercapacitors through the development of composite electrodes that combine various charge storage mechanisms. A new method for preparing composite binder-free MnO_2_/MWCNT/Al electrodes for supercapacitors is proposed. The method is based on the original technique of direct growth of layers of multi-walled carbon nanotubes (MWCNTs) on aluminum foil by the catalytic pyrolysis of ethanol vapor. Binder-free MnO_2_/MWCNT/Al electrodes for electrochemical supercapacitors were obtained by simply treating MWCNT/Al samples with an aqueous solution of KMnO_4_ under mild conditions. The optimal conditions for the preparation of MnO_2_/MWCNT/Al electrodes were found. The treatment of MWCNT/Al samples in a 1% KMnO_4_ aqueous solution for 40 min increased the specific capacitance of the active material of the samples by a factor of 3, up to 100–120 F/g. At the same time, excellent adhesion and electrical contact of the working material to the aluminum substrate were maintained. The properties of the MnO_2_/MWCNT/Al samples were studied by electron probe microanalysis (EPMA), Raman spectroscopy, cyclic voltammetry (CV), and impedance spectroscopy. Excellent charge/discharge characteristics of composite electrodes were demonstrated. The obtained MnO_2_/MWCNT/Al electrodes maintained excellent stability to multiple charge-discharge cycles. After 60,000 CVs, the capacitance loss was less than 20%. Thus, this work opens up new possibilities for using the MWCNT/Al material obtained by direct deposition of carbon nanotubes on aluminum foil for the fabrication of composite binder-free electrodes of supercapacitors.

## 1. Introduction

The development and use of environmentally friendly renewable energy sources is becoming an extremely urgent task all over the world. Over the past decade, there has been tremendous progress in the development of solar energy, hydropower, wind power, tidal energy, and other renewable energy sources [1,2], as well as in technologies of energy harvesting [3]. However, in most cases, clean, renewable energy from these sources cannot be applied directly due to the instability of its generation. Therefore, reliable electrochemical energy storage devices (ECS), including fuel cells, ion batteries, and supercapacitors, are necessary for efficient storage, conversion, and further use of the energy of these sources [4]. Among them, electrochemical capacitors, also known as supercapacitors (SC), are considered a new generation of green energy storage [5]. They have a simple design and are made from cheap, widely used materials, have high power density, fast charging, and longer service life than traditional batteries [6]. Due to this, the areas of application of supercapacitors are extremely wide, from hybrid electric vehicles to miniature current sources in smart things technologies [7,8,9].

In terms of charge storage mechanisms, energy storage/conversion processes for supercapacitors occur in two different ways, which can be generally divided into electrochemical double-layer capacitors (EDLC) and pseudocapacitors [10,11,12]. EDLCs store energy electrostatically by surface adsorption/desorption of ions at the electrode/electrolyte interface, while pseudocapacitors use fast and reversible surface Faraday reactions between an electrolyte and electrochemically active materials [12,13]. CNTs are the most popular one-dimensional nanotubular carbon, which are claimed to have outstanding physical and chemical properties of high electrical conductivity, easily accessible surface areas, good performance chemical stability, and high mechanical strength. These characteristics correspond exactly to the characteristics required by supercapacitors [14,15]. Therefore, in recent decades, CNTs have been widely used either as a material for EDLC electrodes or as conductive supports for pseudocapacitive materials [16,17]. Due to the relatively low specific capacitance of CNTs, they are increasingly being used as the basis for composites with pseudocapacitive materials such as MnO_2_ [18]. Such nanocomposites combine the high pseudocapacitance of MnO_2_ with the high conductivity and mechanical strength of CNTs [19]. In addition, nanocomposites with CNTs have a network structure with open mesopores, which allows electrolyte ions to easily diffuse to the electrode surface [20].

According to the literature data, various methods for the formation of MnO_2_/CNT composites have been developed. In [21], they were obtained by milling the components in an aqueous ammonia solution and subsequent treatment in an autoclave for 5 h at 150 °C. The preparation of a composite by mixing the initial components was also used in [22]. Usually, the chemical reduction of KMnO_4_ in an aqueous solution is used to precipitate MnO_2_. Most often, carbon from CNTs served as a reducing agent [23,24,25,26,27,28,29,30,31]. To reduce permanganate with carbon nanotubes, as a rule, rather harsh conditions are required (treatment for several hours at a high temperature in an autoclave) [26,27,29,31]. However, highly defective CNTs can easily react with a KMnO_4_ solution even at room temperature [32]. Carbon is partially consumed in the reaction with permanganate, which leads to the degradation of CNTs. To avoid this, in [33], the CNT surface was preliminarily covered with a layer of amorphous carbon. Fei Teng et al. [34] precipitated MnO_2_ from MnSO_4_ solution by reaction with a strong oxidizing agent (NH_4_)_2_S_2_O_8_. In [35], solutions of potassium permanganate and Mn(II) acetate were mixed for this purpose. Another way to deposit MnO_2_ on the CNTs surface is the electrochemical oxidation of Mn^2+^ ions [36,37,38,39,40,41]. The characteristics of composites can vary significantly depending on the method and conditions of preparation. Therefore, the results of various works quantitatively vary greatly. However, in all cases, MnO_2_/CNT composite electrodes demonstrate much better electrochemical characteristics compared to electrodes from the original CNTs and MnO_2_ [42,43]. 

An important role is played by the method of manufacturing SC electrodes. Usually, the formation of electrodes from a powdered material requires a binding component that provides contact between the particles of the active material and good adhesion to the conductive substrate [42,43]. In recent years, binder-free electrodes have attracted great interest [44,45,46,47]. Binder-free electrodes have obvious advantages due to the good electrical contact of MnO_2_ with the conductive substrate, improved energy performance due to the exclusion of inactive components, and the possibility of using such electrodes for flexible devices [42]. Roughly two types of binder-free electrodes can be distinguished. The first type is self-supporting structures made entirely of active material. As recent achievements in this direction, recent work on wood-derived thick electrodes for supercapacitors from Nanjing Forestry University should be mentioned. The wood-derived anisotropic structural carbon electrodes have different transfer kinetics of the electrolyte, which significantly affect the electrochemical performance, especially at high current density [44]. A composite wood-derived carbon (WC) electrode (WC@MnO_2_-20) with high-mass-loading of MnO_2_ is reported. Benefiting from the high-mass-loading of the MnO_2_ (≈14.1 mg cm^−2^) and abundant active sites on the surface of the WC hierarchically porous structure, the WC@MnO_2_-20 electrode shows a remarkable high-rate performance of areal/specific capacitance ≈ 1.56 F cm^−2^/45 F g^−1^ even at the high current density of 20 mA cm^−2^ [45]. 

Another type of binder-free electrode is composite structures, in which CNT layers are grown directly on a conductive substrate and then coated with MnO_2_ using one of the above methods. In published articles, buckypaper [28], carbon fiber paper (CFP) and flexible carbon fiber cloth [48], stainless steel mesh [36,37], nickel mesh [24], nickel foam [27], and aluminum foil [49] served as the conductive substrates for such electrodes. It should be noted that aluminum is considered the best metal for a current collector in SC due to its low cost, low specific gravity, excellent electrical conductivity, and high plasticity [50,51,52]. However, there are practically no works on binder-free MnO_2_/CNT/Al electrodes for supercapacitors in the literature. Recently, we have demonstrated a new method for the deposition of MWCNT layers on aluminum foil, which provides excellent adhesion of MWCNTs to the substrate [53]. The resulting MWCNT/Al material showed good performance as SC electrodes. In the present work, we studied the possibility of preparing binder-free MnO_2_/MWCNT/Al composite electrodes using the developed method of MWCNTs grown on aluminum foil.

## 2. Materials and Methods

The aluminum substrate (99.0%, Mikhailovsk, Russia) was a foil 50 µm thick. Strips (0.75 × 3 cm^2^) were preliminarily cleaned with isopropyl alcohol (99.8%, EKOS-1, Staraya Kupavna, Russia) and distilled water. Then they were sonicated for 10 min in a suspension of 45 mg of abrasive powder (28 μm, corundum) in 40 mL of a 20 wt.% Ni(NO_3_)_2_ (99.0%, EKOS-1, Staraya Kupavna, Russia) solution. After that, they were washed with distilled water and placed in a 20 wt.% Ni(NO_3_)_2_ aqueous solution for 20 h for further mild oxidation. Finally, nickel nitrate residues were removed by washing the samples three times with deionized water. The substrates thus prepared after drying in air were used to deposit MWCNTs. The general features of the deposition of MWCNTs on aluminum foil by the catalytic pyrolysis of ethanol vapor are described in a previous article [53]. The masses of the deposited MWCNTs and MnO_2_/MWCNT composite were calculated from the difference in the masses of the substrates before and after synthesis.

To prepare MnO_2_/MWCNT/Al composites, the MWCNT/Al samples were placed in an aqueous solution of KMnO_4_ (high purity grade, Russia) for a certain time at room temperature. Then, the samples were washed several times with deionized water and dried in air for 24 h. In a series of experiments, KMnO_4_ solutions with concentrations of 0.2, 1, and 2 wt.% were used. The duration of treatment varied from 10 to 120 min.

Electrochemical oxidation of MWCNT/Al samples was carried out in a 0.05 M aqueous sodium sulfate solution for 10 min in a two-electrode cell at a potential of 4 V. A counter electrode was a platinum wire. 

A P-40X potentiostat-impedance meter (Electrochemical Instruments, Chernogolovka, Russia) was used for electrochemical measurements. Cyclic voltammetry (CV) and electrochemical impedance spectroscopy (EIS) of the initial and modified samples were carried out in a three-electrode cell. The electrolyte was 0.5 M Na_2_SO_4_ aqueous solution. A saturated calomel electrode (SCE) and platinum wire were used as reference and counter electrodes. Electrochemical impedance experiments were conducted in the presence of 0.5 M aqueous Na_2_SO_4_ solution at a DC potential of 0 V, superimposed by an AC potential of 20 mV peak-to-peak amplitude over a frequency range of 50 kHz to 10 mHz. Scanning electron microscopy (SEM) and the chemical composition of the samples were studied using a JSM 6490 scanning electron microscope (Jeol, Tokyo, Japan) equipped with an INCA Oxford Instruments Electron probe microanalysis (EPMA) system. Raman spectra were recorded using a Sentera Raman microscope (Bruker, Berlin, Germany) under excitation with a solid-state laser with a wavelength of 532 nm. 

## 3. Results and Discussion

### 3.1. Binder-Free MnO_2_/MWCNT/Al Electrodes from As-Prepared MWCNT/Al Samples

As is known, aluminum does not possess the catalytic properties necessary for growing CNTs on its surface by the CVD method. Previously, we developed an original technique for making catalytic activity on an aluminum surface [53]. To deposit MWCNT layers on aluminum foil, we used the catalytic pyrolysis of ethanol vapor. This process does not require complex equipment and proceeds at a relatively low temperature, which is important for an aluminum substrate (Tmp = 660 °C). The features of growing MWCNT layers on aluminum foil are described in detail in a previous article [53]. MWCNT layers grown by this procedure have excellent adhesion to an aluminum substrate and can be used directly as SC electrodes [53]. As noted in the introduction, MWCNTs can serve as the basis for the preparation of MnO_2_/MWCNT composite materials, which have a significantly higher specific electrochemical capacitance compared to the capacitance of MWCNTs due to the pseudocapacitance of MnO_2_. An analysis of the literature shows that the most common way to obtain MnO_2_/MWCNT composites is the direct interaction of carbon nanotubes with a solution of potassium permanganate. The heterogeneous character of the reaction ensures the deposition of the formed MnO_2_ directly on the surface of the nanotubes. According to these reasons, in the present work, we have chosen this simple method to obtain the MnO_2_/MWCNT/Al composite material. An additional argument was the fact that highly defective CNTs quite easily react with KMnO_4_ at room temperature [36]. In previous works, we have shown that the MWCNTs obtained by our method are highly defective [53,54]. The SEM image of the MWCNT/Al composite (Figure 1a) showed a dense fibrous layer on the aluminum substrate. Numerous bends testified to the high defectiveness of MWCNTs. The thickness of the MWCNT layer is about 10 µm. Transmission electron microscopy of MWCNTs (Figure 1b) showed many basal planes (0001) of sp^2^ carbon (“turbostratic structure”). They are visible on the TEM image of the tube (Figure 1b) as a series of parallel lines with an average interval of 0.36 nm. It can be seen that the sp^2^ carbon planes are directed at an angle to the nanotube axis. Thus, the edges of the planes come to the surface of the nanotube, creating a large number of defects [53].

To select the optimal conditions for the preparation of the composite, a preliminary series of experiments was carried out with KMnO_4_ solutions of various concentrations and various processing times. No pretreatment of the MWCNT/Al samples was performed. Next, changes in the elemental composition of the active layer of the MnO_2_/MWCNT/Al samples were compared with changes in specific capacitance and weight gain. Elemental analysis of the surface layer of all treated samples showed the presence of carbon, oxygen, and manganese as the main components. Data on their concentrations are given in Table 1.

As can be seen, the selected mild processing conditions are quite sufficient for the effective interaction of MWCNTs and permanganate. The data obtained by the EPMA method, which is local, should be considered as semi-quantitative. Despite some dispersion of experimental results, the following conclusions can be drawn. With an increase in permanganate concentration, the rate and degree of MWCNT oxidation increase. For 0.2% and 1% solutions, the main increase in the manganese concentration in the active layer occurred during the first 30–40 min. Further treatment in a 0.2% solution even led to a slight decrease in the concentration of manganese, and in a 1% solution, an insignificant gradual increase was observed. In both cases, the oxygen concentration gradually increased. When processed in a 2% solution, an increase in the concentration of manganese and oxygen occurred throughout the entire time. During prolonged treatment (more than 90 min), the layer of MnO_2_/MWCNT was destroyed, and the surface of the aluminum substrate was partly exposed.

Raman spectroscopy confirmed the EPMA data. Figure 2 shows the Raman spectra of the MnO_2_/MWCNT/Al samples obtained after treatment of the initial MWCNT/Al in a KMnO_4_ solution of various concentrations for 90 min. Bands at 1347 and 1604 cm^−1^ belong to MWCNTs (D and G peaks, respectively) [14], and the bands at 505, 580, and 643 cm^−1^ correspond to vibrations of the Mn–O bond [36]. As can be seen from Figure 2, with an increase in the KMnO_4_ concentration, the relative height of the peaks corresponding to manganese increased. After prolonged treatment in a 2% KMnO_4_ solution, their intensity became higher than the intensity of the MWCNTs peaks. 

Cyclic voltammetry showed a significant increase in the capacitance of MWCNT/Al samples after treatment in a KMnO_4_ solution. Figure 3 shows an example CV of samples before and after treatment of various durations. The quasi-rectangular shape of the CV curves demonstrates the good capacitance characteristics of the composite electrodes.

The specific capacitance of MWCNTs and the MnO_2_/MWCNT hybrid material on the electrode was measured according to the generally accepted procedure [51]. The cell capacitance was calculated by Equation (1): *C_cell_* = ∫*IdV*/(∆*V* · *ν*)(1)
where *C_cell_* is the capacitance of the cell, *∫IdV* is the area under the CV characteristic at I > 0 (V · A) (this is half of the area inside the complete loop in Figure 3), ∆*V* is the voltage range (V), and *ν* is the voltage scanning rate (V/s). 

The capacitance of the working electrode in a three-electrode cell is equal to the capacitance of the cell. Therefore, the specific capacitance (*C_spm_*) of active material (MWCNTs or MWCNT/MnO_2_ composite) is *C_cell_/**m*, where *m* is the mass of active material on the electrode. As quantitative characteristics of composite electrodes, we used the values of capacitance per total mass of the working material (*C_spm_*) and capacitance per electrode surface area (*C_sps_*). 

As expected, the composition of the active material and its specific capacitance after treatment of the MWCNT/Al samples in a KMnO_4_ solution depended on both the concentration and duration of treatment. Data on the evolution in the composition of the MnO_2_/MWCNT material, the increase in the mass of the active material, and the increase in specific capacitance depending on the treatment time are collected in Figure 4.

It is believed that the reaction between CNTs and permanganate proceeds according to Equation (2) [24]:4MnO_4_^−^ + 3C + H_2_O = 4MnO_2_ + CO_3_^2−^ + 2HCO_3_^−^(2)

According to this equation, carbon is oxidized to the maximum possible state and removed from the CNT surface. However, it is well known that the strong oxidizing agents (hydrogen peroxide, concentrated nitric and sulfuric acids) react with CNTs to form oxygen-containing functional groups (-C=O, -COOH, etc.) on their surface [14]. Why can a similar process not occur when CNTs are treated with permanganate? Thus, the MnO_2_ deposition may be accompanied by functionalization of the CNT surface. In particular, Chen et al. used a permanganate solution to functionalize CNTs [55]. On the other hand, the oxidation of CNTs also leads to a significant increase in their specific capacitance. This effect is usually associated with the pseudocapacitance of functional groups and an increase in the specific surface area of CNTs due to etching [14]. Thus, an increase in the specific capacitance of MWCNTs after treatment in a KMnO_4_ solution can be associated with both the deposition of MnO_2_ and the oxidation of the MWCNTs surface. Since it is difficult to isolate the contribution of each of the factors, we used the capacitance related to the total mass of MnO_2_/MWCNT material to characterize the samples.

The data in Figure 4 show that, at all permanganate concentrations, the treatment of MWCNT/Al samples in a KMnO_4_ solution led to an increase in the specific capacitance of the samples by several times. The use of 0.2% KMnO_4_ solution gave the least increase in capacitance. For a 2% solution, the increase in capacitance was the largest. However, the value of the specific capacitance of the active material is an important characteristic of the SC electrode, but not the only one. For reliable operation of the MnO_2_/MWCNT/Al binder-free electrode, it is necessary that after its preparation, the aluminum substrate and the MWCNT layer retain the integrity and good adhesion, which ensures good electrical contact between MnO_2_ and the current collector. From this point of view, a 2% solution is less suitable. Considering the above, we have chosen treatment in a 1% KMnO_4_ solution for 40 min as the optimal conditions for MnO_2_ deposition. These conditions were used to prepare MnO_2_/MWCNT/Al samples for further studies. According to the literature, the synthesis of MnO_2_ on CNTs by immersing them in KMnO_4_ solutions has already been published. However, in this work, we had to solve problems related to the activity of MWCNTs synthesized from alcohol vapors. In addition, it was not known how the aluminum substrate would behave in the presence of a strong oxidizing agent. As a result of the experiments performed, the optimal processing conditions were chosen, which make it possible to keep the Al substrate and the MWCNT layer intact and, at the same time, to provide a significant improvement in the characteristics of the electrodes.

In classical supercapacitors, the electrical charge accumulates in an electrical double layer near the electrode’s surface. The process of formation of such a layer is very fast; therefore, SCs are characterized by high charge-discharge rates. The mechanism of charge accumulation in pseudocapacitors is associated with the occurrence of a reversible electrochemical redox reaction [56]. For MnO_2_, Equation (3) describes this mechanism as follows [19]: MnO_2_ + C^+^ + e^−^ = MnOOC(3)

C^+^ (H^+^, Li^+^, Na^+^, K^+^) is an electrolyte cation that is embedded in MnO_2_ on the electrode surface. The process involves a reversible redox reaction between Mn^3+^ and Mn^4+^ [33,41,42,43]. Due to this process, a larger charge can accumulate on the electrode than in a conventional SC. However, this process is slower than the formation of an electrical double layer. In addition, MnO_2_ has a low electrical conductivity, and its deposition can lead to an increase in the ohmic resistance of the electrode, which can also affect the rate of charge/discharge [19,22]. To evaluate the rate of the charge/discharge performance of the obtained binder-free MnO_2_/MWCNT/Al electrodes, a CV study of the dependence of capacitance on the scanning rate was carried out. Figure 5 shows the dependence of the specific capacitance of the active material of the initial MWCNT/Al electrode and the same sample treated in 1% KMnO_4_ solution for 40 min (MnO_2_/MWCNT/Al). 

The *C_spm_* obtained for the MnO_2_/MWCNT/Al electrode was 99.4 F/g at a scan rate of 2 mV/s, which is three times higher than the *C_spm_* obtained for the original MWCNT/Al electrode (33.3 F/g). As the scan rate increased to 100 mV/s, the corresponding capacitance retentions were 88.6% for the original MWCNT/Al electrode and 74.7% for the MnO_2_/MWCNT/Al electrode. Although the retention of the MnO_2_/MWCNT/Al electrode was slightly lower, this result shows an excellent rate capability. As noted above, the mass of the active material of the MnO_2_/MWCNT/Al electrode was about 60% higher than that of the original MWCNT/Al. Accordingly, the observed increase in capacitance per electrode surface area (*C_sps_*) was also greater. The *C_sps_* of the MnO_2_/MWCNT/Al electrode was 22.3 mF/cm^2^ at a scan rate of 2 mV/s, which was five times higher than the *C_sps_* of the original MWCNT/Al electrode (4.4 mF/cm^2^). 

Further study of the electrochemical characteristics of MnO_2_/MWCNT/Al electrodes was carried out using electrochemical impedance spectroscopy. Figure 6a,b show the Nyquist plots of the original MWCNT/Al sample and the MnO_2_/MWCNT/Al sample obtained by its treatment in 1% KMnO_4_ solution for 40 min. Both Nyquist plots are almost linear at low frequencies, indicating good capacitance characteristics (Figure 6a). On the Nyquist plot, the ohmic resistance of a cell is estimated by the offset Z’ along the *x*-axis in the high-frequency region. Accordingly, the ohmic resistance was 4.5 Ω for the original MWCNT/Al sample and 4.6 Ω for the MnO_2_/MWCNT/Al sample (Figure 6b). These results indicate that the deposition of MnO_2_ did not lead to an increase in the ohmic resistance of the cell. Therefore, the MnO_2_ in the MnO_2_/MWCNT/Al composite electrode had good electrical contact with the conductive substrate. At the same time, the slope of the Nyquist plot for the MnO_2_/MWCNT/Al sample in the low-frequency region decreased compared to the initial MWCNT/Al sample (Figure 6a), which can be explained by the pseudocapacitance of MnO_2_. 

From the data of impedance spectroscopy, it was also possible to calculate the frequency dependences of the real and imaginary capacitances (C’(ω) and C”(ω), respectively), which provide valuable information about the characteristics of the SC electrodes. The following Equation (4) was used for these purposes [26]: C’(ω) = −Z”(ω)/(ω · |Z(ω)|^2^); C” (ω) = Z’(ω)/(ω · |Z(ω)|^2^)(4)
where ω is the angular frequency defined as ω = 2πf and |Z(ω)| is the impedance modulus. 

As can be seen from Figure 6c, at a frequency of 0.1 Hz, the real capacitance of the MnO_2_/MWCNT/Al electrode was almost 4.5 times higher than the capacitance of the original MWCNT/Al electrode. This result is in good agreement with the data of cyclic voltammetry (in this case, one should compare the capacitance per electrode area, *C_sps_*). The rate of reversible charge/discharge performance can be estimated from the evolutions of the imaginary capacitance of the MWCNT/Al and MnO_2_/MWCNT/Al electrodes versus the frequency (Figure 6d). The dependences had peaks corresponding to the determined relaxation frequency (fr) and the response time constant τr, defined as 1/fr. The τr is a quantitative parameter to evaluate the fast reversible charge/discharge performance [26]. The MWCNT/Al electrode had a τr of 0.42 s, which showed fast ion transfer capability and excellent rapid charge/discharge performance. The pseudocapacitance restricts the charge/ion rapid transport and leads to a large response time. Therefore, the MnO_2_/MWCNT/Al electrode had a greater value of τr of 3.0 s. However, as applied to pseudocapacitors, this value indicates a very good charge/discharge performance of the obtained composite binder-free electrodes [26].

### 3.2. Binder-Free MnO_2_/MWCNT/Al Electrodes from Oxidized MWCNT/Al Samples

In many works on the preparation of composite MnO_2_/CNT materials for SC electrodes, the surface of initial CNTs was preliminarily oxidized (functionalized). In this work, we also compared the results of the interaction of non-oxidized and oxidized MWCNTs with permanganate solution. As was found earlier, the electrochemical oxidation of MWCNT/Al samples leads to an increase in the concentration of functional oxygen-containing groups on the surface of MWCNTs and an increase in specific capacitance by 4–5 times in the range from –800 to 10 mV and by 1.5–2 times in the range from −10 to 800 mV [54]. At the same time, the electrochemical oxidation of MWCNT/Al somewhat reduces their rate of reversible charge/discharge performance. In the present work, the effect the pre-oxidation has on the properties of MnO_2_/MWCNT/Al binder-free electrodes obtained by treating MWCNT/Al with a KMnO_4_ solution was studied. The initial MWCNT/Al samples were oxidized in a two-electrode cell at 4 V for 10 min in a 0.05 M Na_2_SO_4_ solution. After that, the specific capacitance of MWCNT in the range of −10 to 800 mV increased, on average, by a factor of 1.5. Thus, the starting characteristics of the oxidized MWCNT/Al samples differed significantly from the characteristics of the as-prepared samples. Next, the samples were treated in 1% KMnO_4_ solution for various times. Data on the atomic concentrations of Mn and O, weight increase, and increase in the specific capacitance for the MnO_2_/MWCNT/Al samples thus obtained are shown in Figure 7.

The comparison of these data with data for the MnO_2_/MWCNT/Al samples obtained from as-prepared MWCNT/Al samples (Figure 4e,f,t) shows a greater increase in the concentration of Mn and O in the active layer (Figure 7a), as well as a greater weight gain in the working material (Figure 7b). At the same time, there was no significant improvement in specific capacitance (*C_spm_*) and (*C_sps_*) (Figure 7c, curves 1 and 2, respectively; curves 1’ and 2’ show the growth relative to the oxidized sample).

CVs of the MnO_2_/MWCNT/Al samples prepared from pre-oxidized MWCNT/Al samples showed a similar result. Figure 8a shows the evolution of the CV curves for the same sample: initial (curve 1), electrochemically oxidized (curve 2), and final MnO_2_/MWCNT/Al obtained by treating the oxidized MWCNT/Al sample in 1% KMnO_4_ solution for 40 min (curve 3). It can be seen that the shape and area of the CV loop changed after oxidation. After treatment in a KMnO_4_ solution, a significant increase in the area inside the CV curve was observed; this indicates a significant increase in the electrode capacitance. 

Changes in specific capacitance as a function of scanning rate for the sample after different stages of processing are shown in Figure 8b. At a rate of 2 mV/s, the specific capacitance of the initial sample was 39 F/g; after oxidation, it increased to 64 F/g; after subsequent treatment in a KMnO_4_ solution, it reached 118 F/g. Thus, the specific capacitance of the active material in the final MnO_2_/MWCNT/Al sample increased by a factor of 3 compared to the initial MWCNT/Al sample. With an increase in the scanning rate to 100 mV/s, the specific capacitances of the sample were 36 F/g for the original MWCNT/Al sample, 47 F/g for the oxidized sample, and 90 F/g for the final MnO_2_/MWCNT/Al sample. The capacitance retentions were 92.3%, 73.4%, and 76.3% respectively. These results are close to those obtained by treating as-prepared MWCNT/Al samples with a permanganate solution (see Section 3.1). 

The results of impedance spectroscopy of these samples are shown in Figure 9. At low frequencies, Nyquist plots of initial, oxidized, and final MnO_2_/MWCNT/Al samples were almost linear, indicating good capacitance characteristics (Figure 9a). In the high-frequency region, the Nyquist plots were shifted towards lower Z’ values for the oxidized and final sample compared to the original (Figure 9b). This indicates a decrease in the ohmic resistance of the electrode after electrochemical oxidation from 4.7 to 3.7 Ω. Further deposition of MnO_2_ on the MWCNTs surface did not lead to an increase in the ohmic resistance. The frequency dependence of the real capacitance confirms an increase in the electrode capacitance by about 1.5 times after electrochemical oxidation and by about 4 times after MnO_2_ deposition (Figure 9c). The dependence curves of the imaginary capacitance have maxima, the position of which characterizes the charge-discharge rate. The value of τr for the initial MWCNT/Al electrode was 0.5 s, for the oxidized sample, 0.9 s, and for the final MnO_2_/MWCNT/Al sample, 2 s. Thus, the MnO_2_/MWCNT/Al electrode prepared using intermediate electrochemical oxidation has slightly better charge/discharge performance. However, in general, the characteristics of MnO_2_/MWCNT/Al electrodes obtained from the as-prepared MWCNT/Al samples do not differ much from the characteristics of samples obtained with intermediate electrochemical oxidation.

Thus, the intermediate stage of electrochemical oxidation of MWCNT/Al samples does not significantly improve the characteristics of binder-free MnO_2_/MWCNT/Al electrodes. In addition, the study of the resistance of MnO_2_/MWCNT/Al electrodes to multiple charge-discharge cycles showed significant advantages of electrodes obtained from the as-prepared MWCNT/Al samples. As Figure 10 shows, the MnO_2_/MWCNT/Al electrode prepared from the as-prepared MWCNT/Al sample retained approximately 80% of the specific capacitance after 60,000 charge-discharge cycles. At the same time, the specific capacitance of the electrode obtained with the intermediate stage of electrochemical oxidation of MWCNT/Al dropped to 60% of the initial one after 16,000 cycles. Perhaps this is due to the fact that electrochemical oxidation results in partial etching of the surface of MWCNTs, which makes them less stable. 

For comparison, the as-grown MWCNT/Al electrode after 20,000 charge-discharge cycles in a three-electrode cell almost did not lose capacity. In a two-electrode cell, after 20,000 cycles, the capacity loss was approximately 5% [53]. 

## 4. Conclusions

It has been shown that SC binder-free MnO_2_/MWCNT/Al electrodes can be obtained by a simple treatment in an aqueous solution of KMnO_4_ under mild conditions of MWCNT/Al samples obtained by the direct MWCNTs deposition on aluminum foil. The optimal processing conditions were chosen to ensure a significant increase in the electrode capacitance and at the same time, preserve the MWCNT layer and substrate as much as possible. The treatment of MWCNT/Al samples in a 1% KMnO_4_ aqueous solution for 40 min increased the specific capacitance of the active material of the samples by a factor of three, up to 100–120 F/g. At the same time, excellent adhesion and electrical contact of the working material to the aluminum substrate were maintained. The obtained MnO_2_/MWCNTs/Al electrodes had excellent stability to multiple charge-discharge cycles. After 60,000 CVs, the capacitance loss was less than 20%. The following advantages of the obtained binder-free MnO_2_/MWCNT/Al electrodes can be noted, which make them promising for wide applications. The aluminum substrate has excellent electrical conductivity, low specific gravity, and low cost. The excellent adhesion of the MWCNT layer to the substrate ensures good electrical contact between all electrode components. The obtained electrodes show good capacitance performance and high resistance to multiple discharge-charge cycles. Thus, this work opens up new possibilities for using the MWCNT/Al material obtained by direct deposition of carbon nanotubes on aluminum foil for the fabrication of composite binder-free electrodes of supercapacitors.

## Figures and Tables

**Figure 1 nanomaterials-12-02922-f001:**
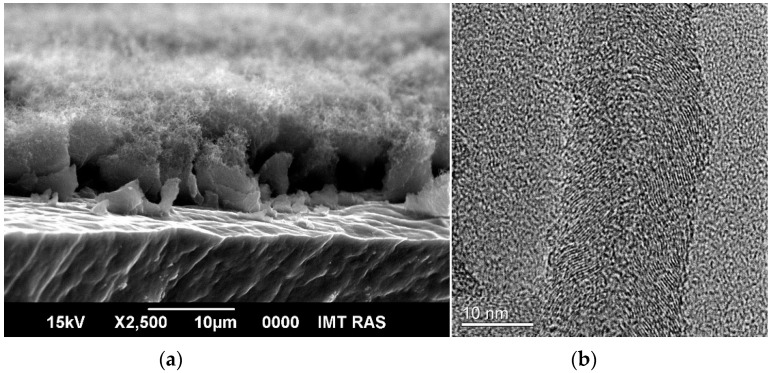
(**a**) Scanning electron micrographs of MWCNT/Al foil: cross-section; (**b**) transmission electron micrograph of individual MWCNT.

**Figure 2 nanomaterials-12-02922-f002:**
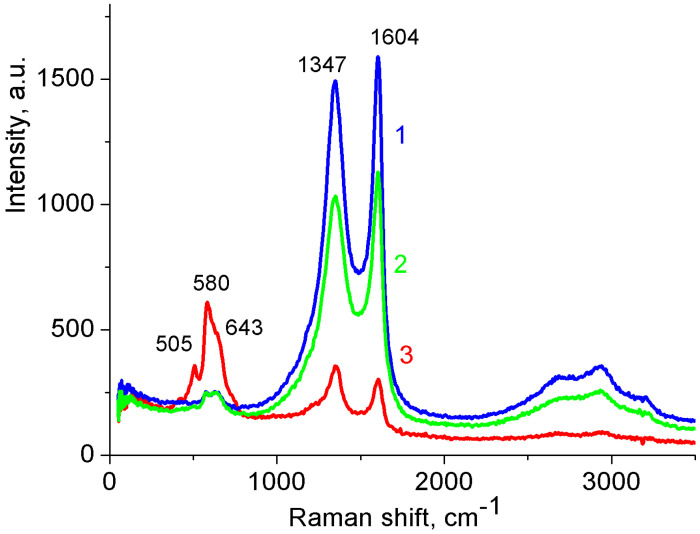
Raman spectra of MnO_2_/MWCNT/Al samples obtained under various conditions. Duration of treatment in KMnO_4_ solution 90 min. Concentration of KMnO_4_ solution, wt.%: 1: 0.2; 2: 1; 3: 2.

**Figure 3 nanomaterials-12-02922-f003:**
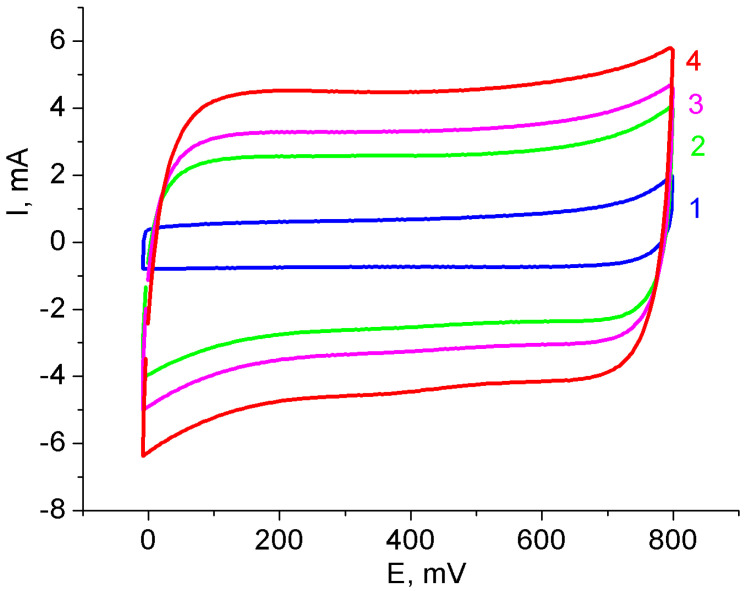
CVs of the MnO_2_/MWCNT/Al foils in the three-electrode cell. Duration of MWCNT/Al treatment in 1 wt.% KMnO_4_ solution, min.: 1: 0; 2: 10; 3: 20; 4: 60. Scanning rate is 100 mV/s.

**Figure 4 nanomaterials-12-02922-f004:**
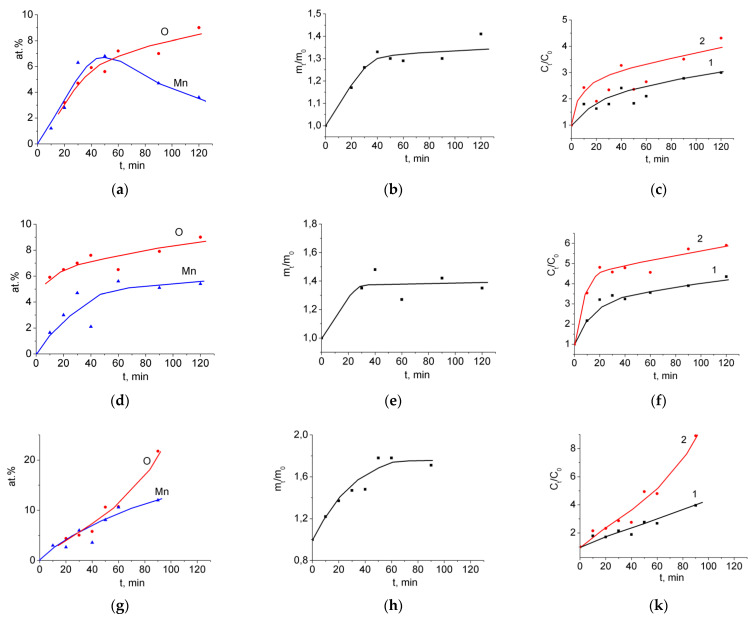
(**a**,**d**,**g**) Concentrations of Mn and O, at.%; (**b**,**e**,**h**) weight increase, m_t_/m_0_; (**c**,**f**,**k**) increase in the specific capacitance, C_t_/C_0_ 1: *C_spm_*, 2: *C_sps_*. Concentration of the KMnO_4_ solution: (**a**–**c**) 0.2 wt.%; (**d**–**f**) 1 wt.%; (**g**–**k**) 2 wt.%.

**Figure 5 nanomaterials-12-02922-f005:**
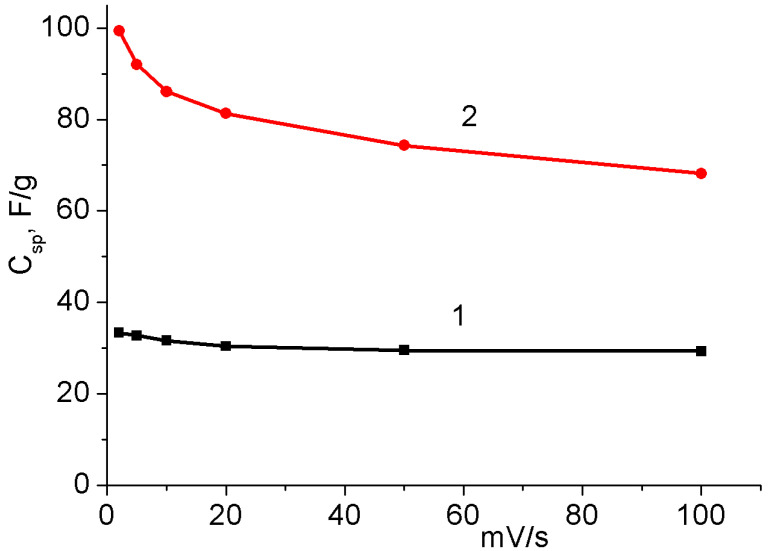
The dependence of the specific capacitance (*C_spm_)* on scanning rate. 1: original MWCNT/Al sample; 2: the same sample after treatment in 1% KMnO_4_ for 40 min (MnO_2_/MWCNT/Al).

**Figure 6 nanomaterials-12-02922-f006:**
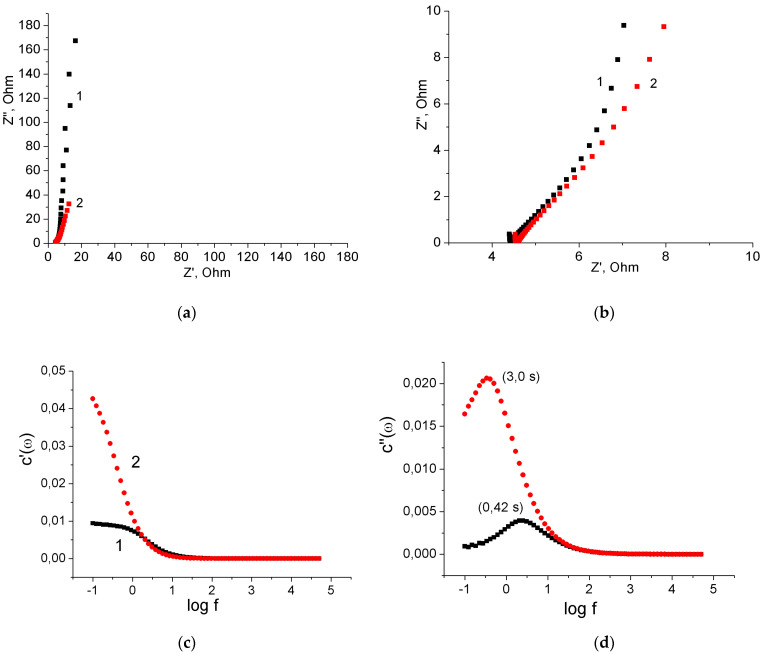
(**a**) The Nyquist plots of the three-electrode cell (full range); (**b**) The Nyquist plots of the three-electrode cell (middle range); (**c**) Frequency dependences of the real capacitances; (**d**) Frequency dependences of the imaginary capacitances. As-prepared MWCNT/Al sample 1, MnO_2_/MWCNT/Al sample 2.

**Figure 7 nanomaterials-12-02922-f007:**
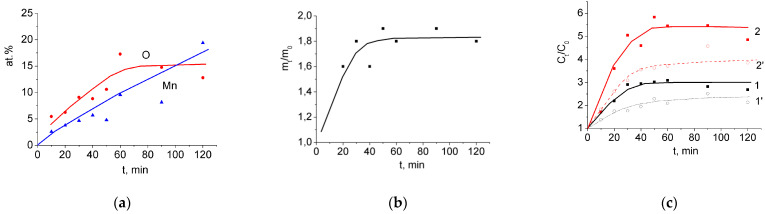
(**a**) Concentrations of Mn and O, at.%; (**b**) weight increase, m_t_/m_0_; (**c**) increase in the specific capacitance, C_t_/C_0_: 1-*C_spm_*, 2-*C_sps_*; curves 1’ and 2’ show the growth relative to the oxidized sample.

**Figure 8 nanomaterials-12-02922-f008:**
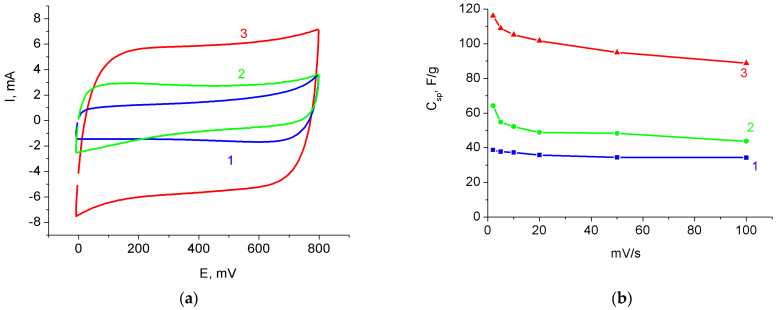
(**a**) CV curves for the original electrode (1), oxidized (2), and treated in permanganate solution (3). (**b**) Dependences of capacities on scanning rate.

**Figure 9 nanomaterials-12-02922-f009:**
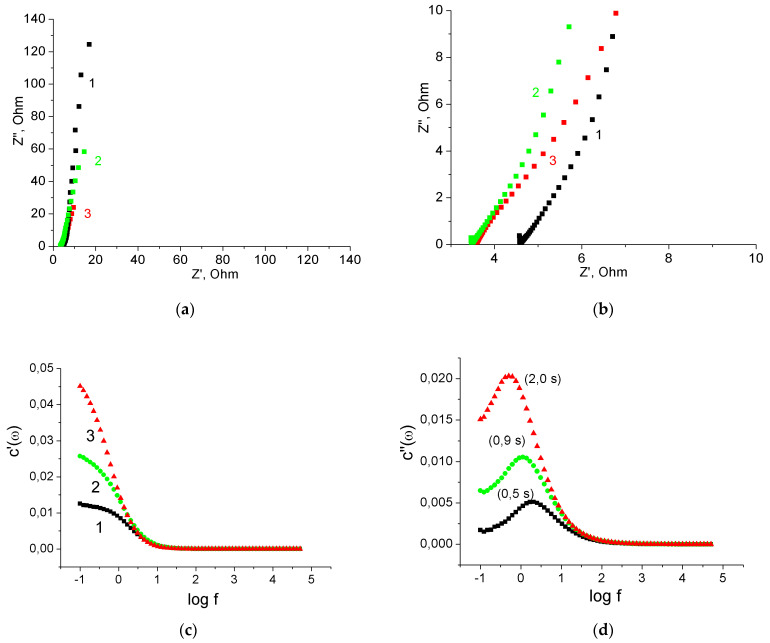
(**a**) The Nyquist plots of the three-electrode cell (full range); (**b**) The Nyquist plots of the three-electrode cell (middle range); (**c**) Frequency dependences of the real capacitances; (**d**) Frequency dependences of the imaginary capacitances. As-prepared MWCNT/Al sample 1, oxidized MWCNT/Al sample 2, MnO_2_/MWCNT/Al sample 3.

**Figure 10 nanomaterials-12-02922-f010:**
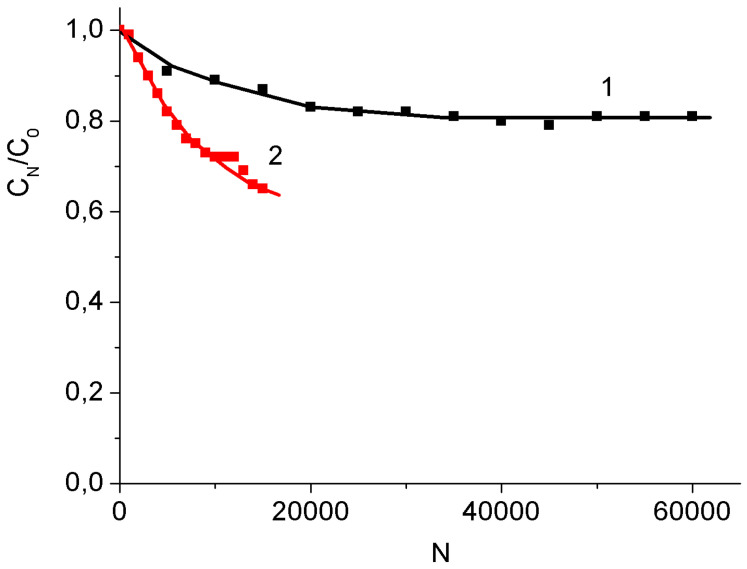
Capacitance evolution during multiple charge-discharge cycling. MnO_2_/MWCNT/Al electrodes from non-oxidized (1) and oxidized (2) MWCNT/Al samples.

**Table 1 nanomaterials-12-02922-t001:** Element concentrations in the active layer of MnO_2_/MWCNT/Al samples depending on the treatment duration and the concentration of the KMnO_4_ solution.

Treatment Duration, min	0.2% KMnO_4_			1% KMnO_4_			2% KMnO_4_		
	C, at.%	O, at.%	Mn, at.%	C, at.%	O, at.%	Mn, at.%	C, at.%	O, at.%	Mn, at.%
10	93	5.8	1.2	92.6	5.8	1.6	-	-	-
20	93.9	3.3	2.8	90.5	6.5	3.0	92.9	4.4	2.7
30	89	4.7	6.3	88.2	7.0	4.8	90.0	4.8	5.2
40	85.5	5.9	8.6	-	-	-	90.6	5.8	3,6
50	87.6	5.6	6.8	-	-	-	81.3	10.6	8.1
60	-	-	-	87.4	7.0	5.6	78.7	10.7	10.6
90	88.3	7	4.7	87.5	7.4	5.1	66.2	21.8	12.0
120	87.4	9	3.6	85.6	9.0	5.4	-	-	-

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
