# Peer review of "Binder-Free MnO2/MWCNT/Al Electrodes for Supercapacitors"

_nanomaterials, 2022, doi:10.3390/nano12172922_

Round 1
Reviewer 1 Report
Redkin et al's paper report the facile synthsis of MnO2/CNT hybrid electrode and their application for supercapacitors. It should be carefully modified before any possible publication.
1. There are some long sentences that should be revised to make the paper more readable. For example, the sentence in the abstract is hard to understand, "It has been shown that a simple treatment of MWCNT/Al samples obtained by this method in an aqueous solution of KMnO4 under mild conditions made it possible to obtain a MnO2/MWCNT/Al composite material perfect for use as binder-free electrodes in electrochemical supercapacitors."
2. In addition to the MnO2/CNT nanocomposites, some metal oxides/CNT hybrid electrodes have been extensively reported, such as NiCo2O4/CNT (Journal of Materials Science & Technology 99 (2022) 260–269), V2O5/CNT(Journal of Colloid and Interface Science 588 (2021) 847–856) and so on. These works should be mentioned in the introduction. As for the carbon material supported MnO2 hybrid electrode, some recently-published papers must be properly cited as well to show the newest progress in this field, such as "Carbon 198 (2022) 46–56", "Carbon 183 (2021) 128-137" and "Renewable and Sustainable Energy Reviews, 158 (2022) 112131".
3. Morphology and phase structure of the MnO2/MWCNT/Al must be characterized to validate the formation of the hybrid materials. Otherwise, I may suspect that the enhancement of the specific capacitance may come from the functionalization of CNTs by KMnO4 treatment.
4. To measure the electrochemical performance of the electrode, GCD tests are more recommended rather than the CV tests.
5. How did the authors evaluate the cycling stability? Generally, it should be tested by GCD method at a certain current density.
Reviewer 2 Report
The manuscript entitled "Binder-free MnO2/MWCNT/Al Electrodes for Supercapacitors" reported a new method for preparing composite binder-free MnO2/MWCNT/Al electrodes for supercapacitors. This work opens up new possibilities for using the MWCNT/Al material obtained by direct deposition of carbon nanotubes on aluminum foil for the fabrication of composite binder-free electrodes of supercapacitor. In general, it is an interesting and valuable topic to deserving a research article.
However, there are still many problems to be solved. So this reviewer would suggest a major revision before its acceptance.
1. One sentence to show the background of the research is suggested at the beginning of abstract.
2. When generally introduce the supercapacitor, some important and recent review article should be carefully read. Please refer: Recent progress in carbon-based materials for supercapacitor electrodes: a review; Design and fabrication of conductive polymer hydrogels and their applications in flexible supercapacitors; etc.
3. One sub-section should be added into section 2 to present the used materials, including company, purity, etc.
4. One image to show the experiment procedure should be added.
5. The morphology of the product is crucial, please provide SEM and TEM images.
6. Why use MnO2 in this work should be further clarified with supporting article: wood‐derived high‐mass‐loading MnO2 composite carbon electrode enabling high energy density and high‐rate supercapacitors.
7. The format of the Figures needs further modification to a clearer version.
8. Equations must be typeset using appropriate fonts and layout. Excel figures must not contain the superfluous outer frame. The same information must not be duplicated in Tables and Figures.
9. The influence caused by the amount of Mn should be further explained, especially more references should be cited for support.
10. Wood-derive carbons are one important kind of binder-free electrodes, which should be mentioned in the last paragraph of introduction. Please refer the recent work on wood-derived thick electrode for supercapacitors from Nanjing Forestry University.
11. Error bars should be added to some of the results of multiple tests to conform to statistical science.
12. The third and fourth parts of the manuscript are lack of literature support, please supplement and upload in the new manuscript.
13. Please draw Ragone diagram and compare the energy density and power density with those of binder-free thick electrodes in previous reports.
14. The authors should pay attention to the difference of “capacity” and “capacitance”. “capacity” is the unit for batteries while “capacitance” is the unit for capacitors. Please refer: Materials & Design 2021, 201, 109518. In addition, the Nyquist plots in Fig. 5a and Fig. 8a were not well displayed. The range for X axial and Y axial should be the same. Please refer: Diamond and Related Materials 2022, 128, 109238.
15. Please carefully check the whole manuscript. There are still some typos and grammar issues. In addition, please carefully check the references to ensure the full information is included.
Reviewer 3 Report
The authors present a work about the synthesis of MnO2-MWCNT/Al supercapacitor electrodes by chemical methods. The MWCNT/Al electrodes are developed through catalytic pyrolysis of ethanol vapor, whereas the Mn oxide material is grown by immersion of the CNTs in KMnO4 aqueous solutions during a controlled time.
The manuscript is well written and the described work represents an interesting method for the development of hybrid electrodes for supercapacitors. However, some additional analyses should be carried out for supporting the conclusions. A major revision of the manuscript must be carried out for reaching the publication standards of Nanomaterials:
1. Throughout the manuscript, the capability of storing energy of the electrode is referred to as “capacity”. However, this term is wrong in terms of supercapacitor devices since it has to be “capacitance”. “Capacity” is related to the stored charge and it is generally used in battery technology. “Capacitance” is the stored charge per unit voltage and it is the figure of merit used in supercapacitors.
2. Introduction:
a. Alternative techniques already developed for the synthesis of MnO2-CNT supercapacitors’ electrodes should be included in the introduction section for comparison (for example: /doi.org/10.1016/j.carbon.2021.08.051; /doi.org/10.1021/acsnano.6b06357; /doi.org/10.1088/1361-6528/aa81b1; /doi.org/10.1021/acsanm.0c02163; /doi:10.3390/ma12030483).
b. The method for the growth of the MWCNT electrodes as well as the synthesis of MnO2 on the CNTs by their immersion of KMnO4 solutions have been already published. The authors must stress the innovation of this work as compared to the literature.
3. The structural properties of the synthesized materials are not studied at all, being of huge influence in the functional properties of the electrodes. The authors should include a structural study of the grown materials (SEM / TEM).
4. The authors assume the presence of only MnO2 phase from the Raman spectra. However, this technique is not sensitive enough for detecting other MnOx phases, which can influence on the capacitance. XPS or SAED analyses would allow identifying the presence of minor phases. Indeed, high resolution C1s XPS analysis would also give key information about the C-O functional groups present in the CNTs and their evolution with the oxidation treatment.
5. The cell capacitance was calculated considering only half of the voltammetry cycle (Eq. 1). Why did not the authors consider the full voltammetry area? This would avoid the influence of asymmetries providing more precise results.
6. Fig 9: It would be very interesting to include the capacitance evolution of the as grown MWCNT/Al and non-oxidized MnO2- MWCNT/Al electrodes.
7. Text issues:
a. Multiwall carbon nanotubes are referred to as “MWCNT” and “MWNT” through the text. Use just one of the abbreviations for coherence.
b. Figs 5d and 8d captions: “Frequency dependences of the IMAGINARY capacitances”.
c. Fig 6 caption: Indicate the sample to which these graphs belong to. Include the description of “1’ ” and “2’ ”.
Round 2
Reviewer 1 Report
Although the authors have made some modification on their manuscript. There are still some issues that were not properly addressed.
1. In the revised manuscript, they provide SEM and TEM images for the CNTs grown on Al foil. But I still insist that they should perform TEM examination on the MnO2/MWCNTs hybrid to clarify the presence of MnO2 on MWCNTs and their size and distribution, which may influence the electrochemical performance noticeably.
2. Some non-English symbols appear in the main text, such as in the 2nd paragraph of P6. Please check the whole manuscript carefully.
3. What levels did they achieve in terms of the electrochemical perfomance, compared with other reports? To demonstrate the advantages of their electrode materials, a table should be added for comparison of the specific capacitance and cycling stability with some previous reports.
Author Response
See attachment please

Reviewer 2 Report
Authors have addressed all the issues well. An acceptance is suggested.
Author Response
We are very grateful for the positive evaluation of the article.
Reviewer 3 Report
- On page 6, there is a paragraph in cyrillic añphabet.
- Fig 7 caption: Indicate the sample to which these graphs belong to. Include the description of “1’ ” and “2’ ” in (c).
Author Response
Many thanks to the reviewer for the very attentive attitude to the manuscript. Corresponding corrections have been made.
- On page 6, there is a paragraph in cyrillic añphabet.
This paragraph accidentally remained in the manuscript after the revision. The bug has been fixed.
- Fig 7 caption: Indicate the sample to which these graphs belong to. Include the description of “1’ ” and “2’ ” in (c).
Description of the curves "1’ " and "2’ " in (Fig.7 c) was included.
Round 3
Reviewer 1 Report
The authors did not provide solid experimental results showing the formation of MnO2 in the composite. Althought raman spectrum shows some charactristic peaks from possible Mn-O bond, we cannot exclude the possibility of other Mn oxides (Mn3O4, MnO), or amorphous MnOx.
Besides, as for the hybrid electrode that they claimed to be new, I cannot agree with them as well. Nanocomposite electrodes like MnO2/CNT have been extensively investigated. Even for the MnO2-loaded CNT arrays, some previous works have been reported (Energy Environ. Sci. 2014, 7: 3709; Nano Lett. 2008, 8: 2664), which show much better electrochemcial performance than this paper. And thus, I cannot find any novelty of this work.
Based on the aforementioned reasons, I cannot recommend this paper for publication in Nanomaterials.
